# Peritoneal Dialysis for Potential Kidney Transplant Recipients: Pride or Prejudice?

**DOI:** 10.3390/medicina58020214

**Published:** 2022-02-01

**Authors:** Luca Nardelli, Antonio Scalamogna, Piergiorgio Messa, Maurizio Gallieni, Roberto Cacciola, Federica Tripodi, Giuseppe Castellano, Evaldo Favi

**Affiliations:** 1Nephrology, Dialysis and Transplantation, Fondazione IRCCS Ca’ Granda Ospedale Maggiore Policlinico, 20122 Milan, Italy; luca.nardelli@unimi.it (L.N.); antonio.scalamogna@policlinico.mi.it (A.S.); piergiorgio.messa@unimi.it (P.M.); fedetr89@gmail.com (F.T.); giuseppe.castellano@unimi.it (G.C.); 2Department of Clinical Sciences and Community Health, University of Milan, 20122 Milan, Italy; 3Department of Biomedical and Clinical Sciences, Università di Milano, 20157 Milan, Italy; maurizio.gallieni@unimi.it; 4Nephrology and Dialysis Unit, ASST Fatebenefratelli Sacco, 20157 Milan, Italy; 5Department of Surgical Sciences, Università di Tor Vergata, 00133 Rome, Italy; rc.1968@icloud.com; 6Kidney Transplantation, Fondazione IRCCS Ca’ Granda Ospedale Maggiore Policlinico, 20122 Milan, Italy

**Keywords:** kidney transplantation, peritoneal dialysis, hemodialysis, patient survival, allograft survival, renal function, delayed graft function, quality of life, outcomes

## Abstract

Kidney transplantation (KT) is recognized as the gold-standard of treatment for patients with end-stage renal disease. Additionally, it has been demonstrated that receiving a pre-emptive KT ensures the best recipient and graft survivals. However, due to an overwhelming discrepancy between the organs available and the patients on the transplant waiting list, the vast majority of transplant candidates require prolonged periods of dialysis before being transplanted. For many years, peritoneal dialysis (PD) and hemodialysis (HD) have been considered competitive renal replacement therapies (RRT). This dualistic vision has recently been questioned by evidence suggesting that an individualized and flexible approach may be more appropriate. In fact, tailored and cleverly planned changes between different RRT modalities, according to the patient’s needs and characteristics, are often needed in order to achieve the best results. While home HD is still under scrutiny in this particular setting, current data seems to favor the use of PD over in-center HD in patients awaiting a KT. In this specific population, the demonstrated advantages of PD are superior quality of life, longer preservation of residual renal function, lower incidence of delayed graft function, better recipient survival, and reduced cost.

## 1. Introduction

Kidney transplantation (KT) represents the “Gold-Standard” of treatment for patients with end-stage renal disease (ESRD) [1,2,3,4,5]. Ideally, a pre-emptive strategy should be adopted to ensure the best results [6,7,8,9]. However, due to the scarcity of donors, the vast majority of transplant candidates require prolonged periods of renal replacement therapy (RRT) before receiving a suitable organ [10,11].

For many years, in-center hemodialysis (ICHD) has represented the only option available [12,13]. In the 1980s, the introduction of peritoneal dialysis (PD) into clinical practice [14,15], raised the question of which RRT should have been preferred in potential KT recipients [16,17,18,19]. Main concerns regarding the routinary use of PD in this particular population were the presence of peritoneal scarring due to previous abdominal surgery, the risk of peri-operative peritonitis [20,21] or exit-site/tunnel infections [20,22], the higher susceptibility to post-transplant diabetes mellitus [23,24], the possible development of encapsulating peritoneal sclerosis [25,26], and the perceived increase in graft thrombosis [23,27,28,29] and acute rejection rates [16,30]. 

Even though several studies have demonstrated that PD does not exert any negative impact on transplant-related outcomes [31,32,33], some clinicians remain reluctant to propose this dialysis option to patients on the transplant waiting list (TWL) [34,35]. Such an attitude is certainly questionable since ICHD, home HD (HHD), and PD should not be regarded as competitive modalities, rather as complementary strategies before [36,37,38,39] and after transplant [40,41,42,43]. Indeed, RRT must be tailored to the specific needs and characteristics of the patient, taking into account the time-dependent variability of these parameters and local facilities [44]. Carefully planned changes between different dialysis techniques can also be considered, in particular circumstances [45,46]. 

While HHD is still under scrutiny in this particular setting [47,48,49,50], there is now evidence suggesting that in patients awaiting a KT, PD may be better than ICHD in terms of quality of life (QoL), residual renal function (RRF) preservation, the incidence of delayed graft function (DGF), graft survival, mortality, and cost.

In the present narrative review, we discuss how the role of PD has changed over time, focusing on the management of patients on the TWL.

## 2. Patient Survival on Renal Replacement Therapy

Patients with ESRD suffer a higher prevalence of cardiovascular disease (CVD), greater incidence of major cardiovascular events, and increased all-cause mortality than the general population [10,51]. These factors can significantly influence the risk of suspension from the TWL and jeopardize recipient and graft survivals after transplant [52].

In a seminal study including 398940 individuals who had started RRT between 1995 and 2000, [53] found that, excluding older diabetic subjects, the adjusted mortality rates on ICHD and PD were substantially similar. Actually, patient survival in both groups varied according to specific clinical characteristics, such as the underlying cause of renal failure, age, and comorbidity [53]. In the same period, a Danish registry analysis, performed on 4568 ICHD and 2443 PD patients, showed that PD could provide a survival advantage over ICHD during the first two years of RRT [54]. Evaluating a Canadian cohort of dialysis patients, Fenton and colleagues also observed that, in young and non-diabetic individuals, PD was associated with higher short-term survival rates than ICHD [55]. A few years later, data collected from the Dutch End-Stage Renal Disease Registry confirmed that patient survival on PD and ICHD was primarily influenced by the presence of diabetic nephropathy and by the age at the time of RRT initiation. In younger (< 50 years) non-diabetic subjects, PD ensured a better overall survival than ICHD. However, the benefit was lost in older (> 60 years) or diabetic individuals [56]. In line with previous studies, a more recent comparison between 6337 pairs of ICHD and PD patients, matched using the propensity score method, showed that individuals starting on PD had an 8% reduction in the overall risk of death compared to those starting on ICHD [57]. 

Over the last two decades, we have witnessed a progressive improvement of dialysis-related outcomes, particularly with home-based modalities [38,49,58,59]. As shown by the US Renal Data System (USRDS) Annual Reports in a population of 650000 subjects on RRT, prevalent patients receiving PD had their survival probabilities doubled from 1996 to 2002 [60] and their adjusted all-cause mortality rates decreased from 164.2 to 131.5 per thousand patient-years from 2009 to 2018 [10]. Such remarkable improvement was evident also among older patients with hypertension or diabetes [10].

## 3. Post-Transplant Recipient and Allograft Survivals

It is demonstrated that KT provides superior life expectancy [61,62,63] and QoL [64,65,66] than dialysis. Nonetheless, returning to dialysis after a failed transplant entails a greater risk of death than starting RRT for the first time [11,40,43,67,68]. Therefore, preserving graft function as much as possible is vital for long-term patient survival. 

Several studies have investigated the impact of pre-transplant dialysis modality on post-transplant outcomes, with mixed results. In the early 1990s, a retrospective analysis on 500 first deceased-donor KT showed no differences between ICHD and PD in short-term patient (88% vs 87%) and graft survival (67% vs 66%) rates [69]. Comparable early patient and graft survivals were also reported in other small series from Ohio State University [70], CHRU Lille [71], and University of Glasgow [72], as well as by a retrospective Medicare database analysis on 22776 transplant recipients [73]. On the contrary, using the USRDS, Goldfarb-Rumyantzev et al. found that pre-transplant PD was associated with a 3% reduction in the risk of graft failure (*p* < 0.05) and a 6% reduction in the risk of death (*p* < 0.001) compared to ICHD [74]. Most of the studies published in the following years and focusing on short- or mid-term outcomes, failed to demonstrate the superiority of an RRT technique over the other ones [75,76,77,78,79]. However, extending the post-transplant follow-up to ten years, Lopez-Oliva et al. managed to show that, despite similar graft survival probabilities (HR = 0.68, 0.41–1.10; *p* = 0.12), recipients previously treated with PD had higher chances of survival than those on ICHD (HR = 2.62, 1.01–6.8; *p* = 0.04) [80]. A better recipient survival (with an equivalent transplant failure rate) was also reported by Schwenger et al. using the large (60008 subjects) database of the International Collaborative Transplant Study Group. On multivariate Cox regression analysis, pre-transplant PD (11664 patients) was associated with a 10% reduction in all-cause mortality (*p* = 0.014) compared to ICHD (45651 patients), with a similar death-censored graft survival rate (*p* = 0.39). Such discrepancy in overall mortality was primarily attributed to a lower incidence of cardiovascular death with preserved graft function observed among PD patients who had received an expanded criteria kidney [81]. Evaluating the Scientific Registry of Transplant Recipients, Molnar et al. found that patients treated with PD before transplant had a lower (21.9/1000 patient-year, 95% CI 18.1–26.5) crude all-cause mortality than recipients previously treated with ICHD (32.8/1000 patient-year, 95% CI 30.8–35.0). More precisely, PD was associated with 43% lower adjusted all-cause mortality, 66% lower adjusted cardiovascular death, and 17% lower unadjusted death-censored graft failure [82]. A recent meta-analysis, including a total of 16 studies published between 1980 and 2014, calculated a pooled adjusted mortality risk ratio of 0.89 (95% CI 0.82–0.97; *p* = 0.006) in favor of pre-transplant PD, with marginal differences in graft survival (pooled adjusted risk ratio 0.97 (95% CI 0.92–1.01; *p* = 0.16) [32]. Analyzing the National Health Insurance (NHI) database, a nationwide cohort study from Taiwan also showed that, in the multivariate analysis, after adjustment for age, sex, time on RRT, and primary renal disease, pre-transplant HD increased the risk of premature transplant loss compared to PD (HR 1.38, *p* < 0.05). Furthermore, higher incidences of new-onset CVD and specific infectious complications such as tuberculosis and hepatitis C were recorded [83]. 

It has been argued that the differences in pre- and post-transplant outcomes between PD and ICHD are due to the fact that patients on PD are generally healthier than their ICHD counterparts [84,85,86,87,88]. In order to address possible bias, several study designs and statistical models have been proposed [89,90]. Among the others, it is worth mentioning the study performed by Kramer and colleagues in 2012, which included 29088 patients from 16 European National or Regional renal registries and used the instrumental variable method to minimize the effect of unmeasured confounders. Standard analysis adjusted for age, sex, primary renal disease, duration of dialysis, donor type, year of transplantation, and country showed that pre-transplant PD was associated with better patient (HR 0.83, 95% CI 0.76–0.91) and graft (HR 0.90, 95% CI 0.84–0.96) survivals than ICHD. However, the instrumental variable model revealed that a 10% increase in the case-mix adjusted center percentage of patients on PD was neither associated with post-transplant recipient survival (HR 1.00, 95% CI 0.97–1.04) nor with graft failure (HR 1.01, 95% CI 0.98–1.04) [91]. 

Overall, current literature demonstrates that pre-transplant PD offers superior or at least equivalent recipient and graft survival rates compared to ICHD [23]. Even though there is evidence that HHD represents a valuable option for many patients with ESRD [49,50,92,93,94], the lack of information regarding KT outcomes in recipients previously treated with HHD does not allow to compare HHD and PD. Due to their methodological limitations, available studies cannot confirm any causality effect between pre-transplant PD and post-transplant outcomes. As a consequence, the exact mechanisms behind the theoretical survival advantage associated with PD remain undetermined. To date, a better RRF at the time of transplant and a reduced incidence of DGF have been recognized as the most plausible contributing factors.

## 4. Delayed Graft Function

Widely accepted definitions of DGF are the need for dialysis during the first week after transplant or a decrease in serum creatinine concentration (SCr) less than 50% from baseline, by post-operative day three (T1/2 SCr) [95,96,97]. DGF is a well-recognized risk factor for peri-operative surgical complications, rejection, and premature transplant loss. Furthermore, the occurrence of DGF can be used as a surrogate marker of late transplant outcomes. In particular, it has been demonstrated that the duration of DGF represents an independent predictor of long-term allograft function and survival [98,99,100,101]. 

The impact of pre-transplant RRT on DGF rate and length has been extensively studied. Back in 1996, Perez-Fontan et al. first evaluated the incidence of DGF in patients who had been treated with PD (n = 92) or ICHD (n = 587) before deceased-donor KT. The proportion of recipients experiencing DGF was 22.5% in the PD group and 39.5% in the ICHD group. Remarkably, dialysis modality was the main predisposing factor for DGF [102]. In a case-control study published in 1999, deceased-donor KT recipients previously treated with PD (n = 117) or ICHD (n = 117) were matched for age, sex, duration of RRT, HLA compatibility, and cold and warm ischemia times. DGF was recorded in 23.1% PD and 50.4% ICHD patients (*p* = 0.0001) with a mean T1/2 SCr of 5.0 ± 6.6 and 9.8 ± 11.5 days, respectively (*p* < 0.0001) [103]. Bleyer and colleagues used the United Network of Organ Sharing (UNOS) database to compare early transplant-related outcomes between PD and ICHD patients. They showed that the odds of not producing urine during the first 24 h after KT were 1.49 (1.28–1.74) times higher in the ICHD group [104]. In the following years, many series and meta-analyses reported results in favor of PD, thus suggesting an association between pre-transplant RRT and DGF [23,29,30,32,33,72,73,82,105]. On the contrary, no significant differences in DGF rates were detected in the studies performed by Caliskan et al. or Dipalma et al. in 2009 and 2016, respectively [31,76].

It has been speculated that the lower incidence of DGF reported among KT recipients previously treated with PD, is actually due to a more favorable peri-operative fluid balance or a better RRF compared to ICHD and HHD, thus reflecting the possibility of indication-related confounders or selection bias [33]. The observation that more than 50% of the transplant candidates on PD had a pre-operative pulmonary arterial pressure (PAP) exceeding 25 mmHg (mean PAP, 21.1 mmHg), certainly supported the hypothesis that PD patients could be frequently over-hydrated (or perhaps under-dialyzed) [106]. However, analyzing data from a cohort of first deceased-donor KT recipients, an elegant study from the University Hospital of Gent, demonstrated that pre-transplant PD and optimized (slightly positive) peri-operative fluid balance were independent predictors of immediate graft function [107]. As a matter of fact, recent data indicate that graft function is more likely affected by intra-operative and early post-operative fluid loads rather than by chronic hydration [108,109]. The RRF at the time of transplant may also play a role [110,111]. Besides, there is now evidence that over-hydration is associated with adverse outcomes among PD and transplant patients [112].

## 5. Residual Renal Function at the Time of Transplant

Many patients with chronic kidney disease (CKD) experience a progressive loss of glomerular filtration rate (GFR) and urinary output. The latter event may eventually lead to decreased bladder capacity, detrusor over-activity, and impaired bladder emptying [113,114,115,116,117]. It is demonstrated that KT recipients with an atrophic or dysfunctional bladder are at higher risk of prolonged catheterization, urinary leakage, and severe vesicoureteral reflux than those with normal bladder volume and function [116,117,118,119,120]. Importantly, dialysis vintage represents the most relevant predisposing factor for irreversible loss of RRF and atrophic bladder [116,118,120,121].

The first report showing that patients on PD maintain a better RRF than those on ICHD was published in 1983 [122]. Since then, a plethora of studies have confirmed the superiority of PD over ICHD or HHD in RRF preservation, with a relative difference in GFR loss ranging from 20% to 80%, depending on the series [123,124,125,126,127,128,129,130]. A very recent study has also shown that PD may slow RRF decline compared to the pre-dialysis [110]. PD can favor RRF preservation through multiple mechanisms. In particular, PD ensures less volume and osmotic pressure fluctuations than ICHD or HHD, thus reducing the occurrence of transient hemodynamic instability. This positive effect seems to be associated with a steadier glomerular capillary pressure and a more consistent glomerular filtration. Moreover, the possibility of avoiding rapid changes in circulating volume and osmolality reduces the risk of renal ischemia. The euvolemic or mild hypervolemic status frequently observed during PD could also contribute to RRF preservation [127,130,131,132,133].

Before transplant, a better RRF has been associated with improved survival in both PD and ICHD/HHD patients [134,135,136,137,138]. The relative contribution of RRF and peritoneal clearance on patient survival during PD has been investigated [127,131,139,140,141]. Among the others, the ADEMEX [139] and the NECOSAD-2 [126] studies have shown that there is an 11% or 12% reduction in the risk of death per 10 L per week per 1.73 m^2^ increment in GFR, without any apparent relationship between survival and PD delivered dose or total small solutes removal. Accordingly, a multivariate logistic regression analysis performed on all Andalusian patients starting PD from 1999 to 2005 (n = 412), has demonstrated that an RRF lower than 4.33 mL/min is an independent predictor of death as much as diabetes, CVD, or age [142].

Further benefits arising from RRF preservation are decreased systemic blood pressure [143,144], reduced left ventricular hypertrophy [145,146,147], increased sodium removal [148,149,150], improved fluid balance [149,151,152], lower serum b2-microglobulin levels [153,154,155], better nutritional status [156,157,158], and lower circulating inflammatory markers [159,160,161,162]. Additionally, RRF facilitates the achievement of adequacy targets [126,163,164,165,166] and may contribute to controlling serum phosphate, uric acid, bicarbonate, and cholesterol levels [161,167,168]. An association between RRF and DGF rate has been established [112,113]. However, the fact that most patients receiving PD exhibit preserved urinary output at the time of dialysis initiation does not allow to rule out possible selection bias.

## 6. Quality of Life on the Transplant Waiting List

KT provides better QoL than dialysis [65,169,170]. Although the average time spent on the TWL varies among countries, it has been estimated that 5% to 15% of the patients remain enlisted long enough to be either removed from the list or die before receiving a suitable organ [52,171,172]. In 2019, among 101000 patients waiting for a KT in the US, more than 8000 died or were suspended due to deteriorating medical conditions, with a median wait-time for a deceased-donor kidney exceeding five years [173]. ESRD and long-term dialysis can affect several aspects of life, negatively impacting physical, psychological, social, or financial wellness. As stated in the campaign “Living Well with Kidney Disease”, ensuring an acceptable QoL during RRT represents a fundamental issue for patients and health care providers [174]. To date, the most used tool for the evaluation of the QoL in patients on RRT is the Kidney Disease Quality of Life (KDQOL) questionnaire [175]. Multiple versions have been proposed, such as the KDQOL Short Form (SF) 1.3, the KDQOL SF 36, and the Short Form 12 [176,177]. The Choices for Healthy Outcomes in Caring for End-Stage Renal Disease (CHOICE) Health Experience Questionnaire is frequently adopted to integrate the SF 36 and it can highlight more granular differences between HD and PD [178].

Compared to ICHD, PD gives the opportunity to dialyze at home, independently or with the help of a caregiver. Additionally, the short amount of time required for fluid exchange allows a flexible schedule and the opportunity to work, travel or participate in recreational activities [179,180,181]. Using the KDQOL SF 1.3, Wakeel et al. compared the QoL of 200 patients on ICHD or PD in Saudi Arabia. Those with cognitive impairment, neurological deficits, or psychiatric disorders were excluded. PD was associated with higher scores in almost all the domains of the questionnaire [182]. De Abreu et al. reported higher degrees of satisfaction and better support from the dialysis staff in PD patients compared to ICHD [183]. In a Brazilian study adopting the KDQOL SF 36 to evaluate 222 ICHD and 116 PD patients, the PD group showed higher scores than the ICHD group in domains related to work status (25 vs 15, *p* = 0.012), encouragement from the dialysis staff (96 vs 83; *p* = 0.008), and patient satisfaction (82 vs 71; *p* < 0.005) [184]. Comparative studies and meta-analyses concluded that PD was associated with less emotional stress [185,186] and a lower odd of cognitive dysfunction [187] than ICHD.

The ability to remain employed after starting RRT is another remarkable aspect of the QoL of dialysis patients [188,189,190]. In this regard, it has been shown that PD offers higher chances of employment than ICHD or HHD [85,191,192].

## 7. Cost of Dialysis before Transplant

Chronic RRT represents one of the most relevant financial burdens for both public and private healthcare systems worldwide. Current projections suggest that the prevalence of ESRD will further increase in the near future due to the rising incidence of diabetes, hypertension, and obesity as well as the progressive aging of the population [193,194,195,196]. It is demonstrated that KT ensures better patient survival and QoL than dialysis, with reduced costs [171,197,198]. Nevertheless, the vast majority of KT candidates spend a considerable amount of time on dialysis before being transplanted [173]. Therefore, the cost of RRT for patients on the TWL should not be neglected.

Over the years, there have been multiple studies aiming to compare the costs associated with PD and ICHD. On the contrary, data on HHD are still scarce [92,93,196,199]. Available literature suggests that ICHD is more expensive than PD, at least in more economically developed countries [200,201]. Nevertheless, mixed results have been reported analyzing the costs of PD and ICHD in Asia [202,203,204,205,206,207] or Africa [208,209,210,211], likely reflecting the impact of geographical, social, and cultural differences in determining the actual expenses related to chronic RRT. Karopadi et al. [212] assessed the cost of PD and ICHD across the world. The results were reported as the annual per patient cost of ICHD divided by the annual per patient cost of PD (ICHD/PD ratio). Forty-six countries were included. The cost of ICHD was 1.25–2.35 higher than PD in 22 countries (17 more economically developed and 5 under-developed), 0.90–1.25 times the cost of PD in 15 countries (2 more economically developed and 13 under-developed), and 0.22–0.90 times the cost of PD in 9 countries (1 more economically developed and 8 under-developed). Overall, these data confirm that, in more economically developed countries, PD is less expensive than ICHD. In less economically developed countries, PD is a financially suitable option provided that an economy of scale is achieved, with local production or low import duties on dialysis equipment [212].

According to the USRDS 2020 Annual Data Report, total inflation-adjusted Medicare expenditures per patient with ESRD rose, between 2009 and 2018, by more than 2% (from 40.9 to 49.2 USD billion). ICHD remained the most expensive RRT at 93191 USD per person annually, whereas per person per year spending for those receiving a KT or remaining on PD was 37304 and 78741 USD, respectively [10]. It can be argued that such a remarkable difference between ICHD and PD is due to a selection bias, as sicker and, therefore, more costly patients are referred to ICHD. Furthermore, the costs emerging from PD failure and the shift to ICHD should be considered. However, an annual per patient saving of about 15000 USD and reduced costs for patients switching from PD to ICHD compared to those remaining on ICHD have been reported [213,214]. Certainly, it would be relevant to evaluate, with properly designed economic health-related studies, the costs of hospital admissions and hospital attendances for vascular accesses vs PD catheters procedures and complications [215].

## 8. Conclusions

Historically, ICHD has been the preferred RRT modality for most ESRD patients awaiting a KT. Over the years, several studies have demonstrated that the perceived advantages of ICHD over PD, in this particular population, are not supported by solid evidence. On the contrary, a critical analysis of the literature seems to suggest that PD may be particularly beneficial for transplant candidates. Demonstrably, patients on the TWL who receive PD have reduced all-cause mortality, improved QoL, and reduced costs compared to those on ICHD. Also, properly delivered PD allows avoiding vascular access placement and fistula-related cardiocirculatory overload. After transplant, recipients who were on PD benefit from lower incidences of DGF and fewer peri-operative urological complications. However, further and properly designed studies are needed to confirm that these encouraging results eventually translate into better long-term graft or recipient survival. More comparative data on HHD is also warranted.

## Data Availability

Not applicable.

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
