# Peer review of "Peritoneal Dialysis for Potential Kidney Transplant Recipients: Pride or Prejudice?"

_medicina, 2022, doi:10.3390/medicina58020214_

Round 1
Reviewer 1 Report
-The authors did not explain why the graft and patient outcomes were significantly better among those who received pretransplant PD vs. HD. If they considered that RRF at the Time of Transplant might be the cause, so the subtitle should be modified.
Author Response
Reviewer 1
We are grateful to Reviewer 1 for the positive feedback and the valuable comments provided during the evaluation of the manuscipt.
Q1) The authors did not explain why the graft and patient outcomes were significantly better among those who received pretransplant PD vs. HD.
A1) To date, there is enough evidence to support the hypothesis that pre-transplant PD is associated with improved recipient survival after kidney transplantation. On the contrary, data demonstrating a possible association between pre-transplant PD and improved graft survival are unconvincing. Due to serious methodological issues (poor quality design, selection bias, confounding by indication, and limited follow-up), available studies cannot confirm a direct causality between pre-transplant PD (or preserved RRF at the time of transplant) and better recipient survival. In order to explain this lack of information (or missing link), in the revised version of the manuscript, we have included the following paragraph: “Overall, current literature demonstrates that pre-transplant PD offers superior or at least equivalent recipient and graft survival rates compared to ICHD. Due to their methodological limitations, available studies cannot confirm any causality effect between pre-transplant PD and post-transplant outcomes. As a consequence, the exact mechanisms behind the theoretical survival advantage associated with PD remain undetermined. To date, a better RRF at the time of transplant and a reduced incidence of DGF have been recognized as the most plausible contributing factors.
Q2) If they considered that RRF at the Time of Transplant might be the cause, so the subtitle should be modified.
A2) As stated above, current literature suggests an association between RRF at the time of transplant and post-transplant outcome. However, no direct causality has been demonstrated. As such, it sounds more reasonable to consider RRF as a possible contributing factor.
Kindest regards,
Evaldo Favi
Reviewer 2 Report
Concerns:
Manuscript in Medicine (MDPI)
In the manuscript:” Peritoneal Dialysis in Potential Kidney Transplant Recipients: 2 Pride or Prejudice?” the authors authors presented a review of the literature on renal replacement therapy in patients with chronic kidney disease.
The main idea of ​​the study was to evaluate peritoneal dialysis (PD) as one of the options, in addition to hemodialysis (HD), for treatment before kidnet transplantation (KT), and to assess which method is more optimal for patients planned for kidney transplantation.
The paper is generally well written , the aim of this study is clear. The authors chose the appropriate literature.
Strengths of the article
The authors, based on numerous cohort studies, indicated many commonly known facts: :
- The authors point out that patients with end stage renal disease suffer higher prevalence of cardiovascular disease; greater incidence of major cardiovascular events, and increased all-cause mortality than the general population.
- Demonstrated the benefits of choosing PD for transplant patients submitted to the transplant waiting list for transplant patients.
- Confirmed the benefits of longer maintenance of residual diuresis in patients treated with PD compared to HD.
- They cited works proving a better course after KT treated in the pre-transplant period with PD.
- Until 1980, they recalled, HD was the only available dialysis treatment option.
- They cite literature suggesting that in patients awaiting a KT, PD may be better than HD in terms of quality of life, incidence of delayed graft function, post-transplant allograft survival, and cost.
- PD allows avoiding to violate the cardio-vascular system and less peri- operative urological complications.
Weaknesses of the article
- All the facts presented are commonly known and obvious.
- Out of 131 items in the literature, only 8 items are from 2015 and above; current items have not been used.
- The ratio of peritoneal dialysis to hemodialysis is still 1-2 to 9-8 worldwide, which is due, inter alia, to the choice of patients and also medically. So, despite the undeniable benefits of choosing peritoneal dialysis as the first option for renal replacement therapy for patients, the doctor does not have a large choice of dialysis method.
In summary, a question arises what is new about the work.
In my opinion paper is not of sufficient novelty to be published.
Author Response
Reviewer 2
We are grateful to Reviewer 2 for evaluating the manuscript. We also appreciate the comments provided, accepting his/her personal points of view regarding the suitability and novelty of our work.
Q1) In the manuscript:” Peritoneal Dialysis in Potential Kidney Transplant Recipients: 2 Pride or Prejudice?” the authors authors presented a review of the literature on renal replacement therapy in patients with chronic kidney disease. The main idea of ​​the study was to evaluate peritoneal dialysis (PD) as one of the options, in addition to hemodialysis (HD), for treatment before kidney transplantation (KT), and to assess which method is more optimal for patients planned for kidney transplantation. The paper is generally well written, the aim of this study is clear. The authors chose the appropriate literature.
Strengths of the article
- The authors, based on numerous cohort studies, indicated many commonly known facts:
- The authors point out that patients with end stage renal disease suffer higher prevalence of cardiovascular disease; greater incidence of major cardiovascular events, and increased all-cause mortality than the general population.
- Demonstrated the benefits of choosing PD for transplant patients submitted to the transplant waiting list for transplant patients.
- Confirmed the benefits of longer maintenance of residual diuresis in patients treated with PD compared to HD.
- They cited works proving a better course after KT treated in the pre-transplant period with PD.
- Until 1980, they recalled, HD was the only available dialysis treatment option.
- They cite literature suggesting that in patients awaiting a KT, PD may be better than HD in terms of quality of life, incidence of delayed graft function, post-transplant allograft survival, and cost.
- PD allows avoiding to violate the cardio-vascular system and less peri- operative urological complications.
A1) We would like to thank Reviewer 2 for recognizing the strengths of the paper.
Q2a) All the facts presented are commonly known and obvious.
A2a) We accept the criticism. Nevertheless, the fact that the vast majority of patients with ESRD remain on ICHD rather than home dialysis mostly reflects, according to several studies, a generalized lack of information and a sort of prejudice toward PD or HHD (among patients, caregivers, and clinicians).
Q2b) Out of 131 items in the literature, only 8 items are from 2015 and above; current items have not been used.
A2b) We agree with Reviewer 2. Accordingly, in the revised version of the manuscript, there are more than 50% of the references from 2015 or above.
Q2c) The ratio of peritoneal dialysis to hemodialysis is still 1-2 to 9-8% worldwide, which is due, inter alia, to the choice of patients and also medically. So, despite the undeniable benefits of choosing peritoneal dialysis as the first option for renal replacement therapy for patients, the doctor does not have a large choice of dialysis method.
A2c) While respecting the personal point of view of the Reviewer, we think that the small proportion of patients on home dialysis (PD or HHD) mostly reflects a lack of personalized counselling, training, and care.
Kindest regards,
Evaldo Favi
Reviewer 3 Report
This is a very nice summary of the PD vs. HD. Although, the authors tone was obviously favoring PD in the entire manuscript. There are a few surmountable limitations.
- DGF may be a function of RRF, which may be the reason a person was recommended PD (confounding by indication). less observed DGF is confounded by the modality of RRT (selection bias)
- There is no mention of the home hemodialysis modality in the entire manuscript. I encourage the authors to comment on any studies or address it as a limitation.
- In the conclusion section "violating cardiovascular" PD can do this with inadequate dialysis.
- Minor edits:
- please change in waiting list to on waiting list
- Fourth-six should be forty-six
Author Response
Reviewer 3
We are grateful to Reviewer 3 for the positive feedback and the valuable comments.
Q1) This is a very nice summary of the PD vs. HD. Although, the authors tone was obviously favoring PD in the entire manuscript. There are a few surmountable limitations.
A1) Thank you very much for your appreciation. As declared in the cover letter, the overall tone of the review if quite provocative and clearly in favor of PD.
Q2) DGF may be a function of RRF, which may be the reason a person was recommended PD (confounding by indication). Less observed DGF is confounded by the modality of RRT (selection bias).
A2) We agree with Reviewer 3. Accordingly, we have discussed the possibility of confounding by indication or selection bias in the sections related to RRF and DGF. Also, we have clearly stated that current studies cannot demonstrate any direct causality between pre-transplant PD (or RRF at the time of transplant) and reduced DGF or premature graft loss rates.
Q3) There is no mention of the home hemodialysis modality in the entire manuscript. I encourage the authors to comment on any studies or address it as a limitation.
A3) As suggested by Reviewer 3, in the revised version of the manuscript, we have separately discussed the literature related to ICHD and HHD. Also, we have highlighted the fact that current data do not allow a proper comparison between KT-related outcomes of recipients previously exposed to PD or HHD.
Q4) In the conclusion section, "violating cardiovascular" PD can do this with inadequate dialysis.
A4) We agree with Reviewer 3. Accordingly, we have rephrased the sentence: “Also, properly delivered PD allows avoiding to violate the cardio-vascular system through vascular access placement and cardiocirculatory overload.
Q5) Please change in waiting list to on waiting list.
A5) Done.
Q6) Fourth-six should be forty-six
A6) Done.
Kindest regards,
Evaldo Favi
Round 2
Reviewer 2 Report
Dear Editor
Concern:Manuscript ID: medicina-1534105
Taking into account the review of the second reviewer and the explanations of the authors, I accept the publication of the manuscript.
Kind regard
Reviever 1
Katarzyna Madziarska